# Projected Status of the Ghost Orchid (*Dendrophylax lindenii*) in Florida during the Next Decade Based on Temporal Dynamic Studies Spanning Six Years

**DOI:** 10.3390/plants10081579

**Published:** 2021-07-31

**Authors:** Ernesto B. Mújica, Adam R. Herdman, Mark W. Danaher, Elaine H. González, Lawrence W. Zettler

**Affiliations:** 1Orquideario Soroa, Carretera a Soroa Km. 8, Candelaria 22700, Cuba; emujica@uart.edu.cu (E.B.M.); egh@uart.edu.cu (E.H.G.); 2Department of Biology, Southern Illinois University-Edwardsville, Edwardsville, IL 62026, USA; aherdma@siue.edu; 3Florida Panther National Wildlife Refuge, U.S. Fish & Wildlife Service, Immokalee, FL 34142, USA; mark_danaher@fws.gov; 4Department of Biology, Illinois College, 1101 W College Ave., Jacksonville, IL 62026, USA

**Keywords:** hydrology, conservation, seedling recruitment, senile populations, species recovery

## Abstract

The enigmatic ghost orchid, *Dendrophylax lindenii* (Lindley) Bentham ex Rolfe, is a showy leafless epiphyte restricted to low-lying forests in south Florida and western Cuba. Because of its appeal and reputation for being difficult to cultivate, *D. lindenii* remains vulnerable to poaching and environmental changes. About 2000 individuals are assumed to remain in Florida, most confined within water-filled cypress domes in the Fakahatchee Strand, but virtually no information exists on current population numbers throughout the region. This paper provides a preliminary summary of the ghost orchid’s projected status based on six continuous years of data collected within the Florida Panther National Wildlife Refuge (FPNWR) from 2015–2020. The orchids were clustered in seven different populations, each separated by ca. 5 km. Quantitative data were collected spanning three age classes (seedlings, juveniles, mature plants) for each population, and survival, flowering, and fruiting were noted. To estimate the temporal variability in the demographic rates, size-structured integral projection models (IPMs) were constructed for each annual transition (e.g., 2015–2016, 2016–2017). Results for all seven populations pooled suggest that *D. lindenii* numbers will decline by 20% during the next decade in the absence of external adverse factors. Seedling recruitment is not expected to keep pace with the projected decline. Only one population, which was also from the wettest location, continuously harbored spontaneous seedlings, suggesting that most populations within the FPNWR lack conditions suitable for reproduction.

## 1. Introduction

The ghost orchid, *Dendrophylax lindenii* (Lindley) Bentham ex Rolfe, is one of the rarest and most enigmatic orchids in the Western Hemisphere. Restricted to low-lying forests in southern Florida and western Cuba, the species exists as leafless epiphyte throughout most of its life, growing abaxially on the host tree (Figure 1) [1], typically on corrugated bark surfaces with a northern orientation [2]. The species is easily overlooked due to the gray–green coloration of the photosynthetic roots and lack of leaves until it yields a striking floral display consisting of 1–4 long-lasting, fragrant, white flowers from May to December (Figure 1) [2]. Because of its appeal and reputation for being difficult to cultivate, *D. lindenii* remains vulnerable to poaching. Other threats to the species’ long-term survival include phytophagous pests [3], hydrological changes to its low-lying wetland habitat [4,5], and periodic hurricanes [6], in addition to climate change [7].

There are currently no estimates of the future status of the ghost orchid in Florida. The current number of individuals is assumed to be less than 2000, mostly confined to remote eastern Collier County within a narrow (50 × 100 km) corridor protected by the Florida Panther National Wildlife Refuge (FPNWR) and Fakahatchee Strand Preserve State Park to the south (Figure 1). In Cuba, fewer individuals (<500) are thought to remain, most on the westernmost tip within Guanahacabibes National Park. Contrary to earlier reports (e.g., [8]), *D. lindenii* is not known to occur in the Bahamas [9]. Although *D. lindenii* is listed as endangered by the State of Florida [5], it is not Federally-listed because it is also found outside of the United States (Cuba). This assumption has yet to be substantiated by much-needed genetic comparisons [10]. Given that ghost orchids in Cuba differ in having slightly larger flowers that open later in the year (October to December) [2], the possibility remains that *D. lindenii* populations in Florida differ taxonomically from those in Cuba.

To conserve orchids vulnerable to extinction, such as *D. lindenii*, high quality, long-term monitoring data are needed to provide accurate forecasting of extinction risks [11,12]. This is best achieved by collecting data annually and linking it with drivers of plant population dynamics [11]. Despite a wealth of information on orchids in the literature dating back over a century (e.g., [13]), few studies have documented and analyzed orchid species declines through time, exposing a serious knowledge gap. This lack of information comes at a critical time when natural habitats remain under siege by human impacts (e.g., climate change, deforestation).

Because the ghost orchid is rare, geographically confined everywhere it grows, and appears to have different population dynamics relative to other epiphytic orchids, studies are urgently needed that elucidate the number of existing plants and how many there will be, in light of rapid environmental changes taking place throughout the region. Information must also be obtained describing *D. lindenii*’s specific habitat requirements, and the ecological components needed for its successful reproduction from seed (e.g., pollinators, mycorrhizal fungi). In 2015, we began a joint research collaboration between the United States and Cuba aimed at collecting and sharing yearly survey data with the common goal of conserving *D. lindenii* in both countries. This paper provides a preliminary summary of the ghost orchid’s status in Florida based on six continuous years of data collected within the Florida Panther National Wildlife Refuge from 2015–2020. We conclude this paper with recommendations that have the potential to benefit the species’ long-term survival in south Florida based on the data.

## 2. Results

### 2.1. Orchid Numbers

Prior to 2015, 16 *D. lindenii* individuals were known to occur within the Florida Panther National Wildlife Refuge scattered throughout the 10,684 ha area but confined to five different cypress domes (populations). Our efforts to document additional orchids over a six-year period revealed hundreds of individuals spanning different growth stages (Table 1), with a total of 652 individuals having been recorded as of 2020. Two new populations were also documented, raising the total number to seven, all consisting of deep-flooded forested wetlands with a persistent hydroperiod (Figure 2). No ghost orchids were documented from any of the dryer surrounding areas despite numerous attempts to locate the species throughout the refuge during all six years. Surveys in the same sites during 2016 and 2017 revealed the presence of 72 additional orchids in higher growth stages that were overlooked in 2015, and they were added to the total from 2015. With each year of careful monitoring combined with previous experience at spotting this leafless species, the total number of *D. lindenii* recorded in the refuge continued to rise. Of the orchids that were initially documented in 2015 and monitored through all years thereafter (Table 1), the number of mature plants at the FPNWR decreased by 48 from 2017 to 2018. During the 2018 survey, we noted several host trees and branches harboring *D. lindenii* that were toppled by high winds (likely due to Hurricane Irma on 10 September 2017), which contributed to the mortality of mature and juvenile orchids recorded. Seedling numbers also dropped sharply after the hurricane. For example, prior to the cyclone, 155 seedlings were recorded in 2017, but seedling numbers dropped thereafter (2018 = 130, 2019 = 55; Table 1). In addition to high winds, heavy rainfall from the 2017 hurricane also likely contributed to mortality, as several orchids affixed to the lowest portion of the host tree’s main trunk at the point just above the waterline were recorded as dead in subsequent years. During 2017—the wettest year on record for Collier Co. (256.5 cm)—the FPNWR received 78% of the year’s annual rainfall between 1 July and 30 September, which resulted in orchids lower on the host trees to be completely submerged for a prolonged period of time which we attribute to the mortality observed.

### 2.2. Fecundity

The total number of ghost orchid inflorescences observed within the FPNWR each year was fewer than 100, with the highest number (92) recorded in 2019 (Table 1). Most of the orchids that flowered yielded a solitary inflorescence with 1–2 flowers (Table 1). Each flower remained receptive to pollination for ca. 3 weeks after fully opening; however, shortly (24 h) after being pollinated (evidenced by the removal of pollinia and/or darkened stigmatic surface), each flower changed in appearance from being white to translucent, and displayed noticeable withering leading to fruit set. Only 2% of 7% of all flowers developed fruits in a given year, with one exception (2018), when 21 of 123 flowers (17.1%) yielded capsules. A similar percentage was observed the following year (2019) when 16 of 96 (16.7%) flowers developed fruits (Table 1).

### 2.3. Demographic Patterns and Population Growth

Based on our AICc models, annual differences were noted in yearly demographic patterns of *D. lindenii* spanning all six years for the seven populations pooled (Figure 3). Lambda (λ) values revealed an increase (λ > 1) in the number of ghost orchids in the first two years of the survey (Figure 3), but a noticeable decrease in the third year (λ = 0.8939; Table 1). The population began to recover in the last three years, but λ values remained lower than 1 (Figure 3).

When λ values for each of the seven populations were analyzed separately, Population A was the most stable, with λ values closer to 1 but still depicting a decline (Figure 4). We attribute this stability to the high number of seedlings recorded at the site, which were noticeably lacking in the other six populations surveyed with one exception: Population D with 15 seedlings or 23.3% of the total for 2019 (Table 1). When data were pooled for all seven populations for all six years (Figure 5), *D. lindenii* numbers are projected to drop by ca. 20% within the FPNWR during the next 10 years in the absence of external adverse factors (i.e., hurricanes, poaching, climate change). Population A is projected to be the most stable among the seven sites during the next 10 years (Figure 6), supported by the high seedling recruitment observed. However, the number of mature plants (C-3, C-4) at this site will experience a steady decline during the next 10 years based on the projections (Figure 7). Consequently, the drop in the number of orchids capable of reproduction would be expected to negatively impact the number of spontaneous seedlings generated in the years beyond 2030. When all populations within the refuge are combined, taking into account the different life stages, there is a marked decrease in the number of mature orchids (C-3, C-4) projected during the next decade, although seedling recruitment would remain relatively stable (Figure 8). However, these seedling totals were elevated by data obtained from Population A. Thus, seedling recruitment projections during the next decade (Figure 8) may be too optimistic if Population A is not factored in. Mortality could also be artificially inflated among age distributions in cases where clusters of one orchid life stage were recorded as dead following the death of the host tree.

## 3. Discussion

This study reports the first six years of detailed survey data collected annually in the Florida Panther National Wildlife Refuge involving a collaborative research project between the United States and Cuba that is expected to last 10–20 years. As such, the results should be regarded as preliminary. Nevertheless, this study provides the first quantitative evidence that the endangered and highly coveted ghost orchid, *D. lindenii*, is experiencing a decline in overall numbers within the Fakahatchee Strand. This small, narrow area in south Florida is thought to harbor the largest remaining number of ghost orchids and also the highest diversity of orchids in North America, excluding Mexico [8,14,15]. During the next decade (2020–2030), overall ghost orchid numbers are projected to drop by 20%, and this projection excludes the possibility of adverse external factors (e.g., hurricanes) that could accelerate the decline. This was evident in our survey when we noted a drop in 48 ghost orchid numbers from 2017 to 2018, which we attribute to Hurricane Irma that made landfall on 10 September 2017 at Marco Island ca. 50 km from the SE of the FPNWR. This study also revealed that only one of the seven sites (Population A) harbored spontaneous seedlings during all six years when data were collected, and this site was also the wettest, i.e., it maintains relatively high water levels throughout the year [10]. One site (Population D) harbored 15 seedlings that were recorded in the 2019 survey, but these seedlings did not survive the following year. Accordingly, six of the seven populations within the FPNWR appear to be “senile” (i.e., lacking conditions suitable for reproduction). Rasmussen et al. [16] characterized orchid populations as “senile” when they consist of mostly mature plants in habitats lacking critical environmental conditions necessary for seedling recruitment. Although it is still too early to assume that these six populations have earned this designation, this survey serves a useful purpose by providing land managers with an early warning mechanism for identifying the most vulnerable populations, instilling confidence in making decisions aimed at long-term planning and management. It also underpins the need to collect more information on the factors required for successful seedling recruitment in these areas.

### 3.1. Water, Mycorrhizal Fungi, and Pollinators

What we currently know about *D. lindenii* and Population A may reveal new insights into why this population continues to generate spontaneous seedlings. High moisture levels at this location often extend into the dry season (November to March) when occasional subfreezing temperatures and low relative humidity levels occur. Thus, the orchids affixed to host tree branches closer to the water level in forest “domes” may be insulated from these adverse seasonal conditions that would otherwise be lethal, as implied by Luer [8]. This may be especially true for seedlings that have yet to form mature roots that cover and protect the sensitive shoot region on the exposed surface [1]. Water availability is also critical for the survival and persistence of free-living saprotrophic fungi that decompose organic matter [17], such as the mycorrhizal associates of *D. lindenii*. Johnson [18], Hoang et al. [1], and Mujica et al. [2] all revealed that *D. lindenii* associates with basidiomycetes in the genus *Ceratobasidium* D.P. Rogers. Using modern molecular techniques, Johnson [18] determined that *D. lindenii* is a fungal specialist targeting a single clade of *Ceratobasidium* representing a rare OTU (operational taxonomic unit). Curiously, this fungus was widespread on trees in *D. lindenii* habitats throughout the FPNWR [18], yet most seedlings recorded in our study were restricted to one site (Population A). This would suggest that some other factor may be absent in these other sites, and we suspect the missing component may be linked to lower water levels.

Available moisture at Population A also may affect the hawk moth pollinators of *D. lindenii* by reducing the distribution of potential host plants needed by their larvae. Danaher et al. [10] documented at least two hawk moth pollinators (Lepidoptera: Sphingidae) of *D. lindenii* in the FPNWR (*Dolba hyloeus*, *Pachylia ficus*) and four other species are suspected of being capable of pollination. Larvae of both *D. hyloeus* and *P. ficus* are reported to feed on *Ficus* and *Ilex* trees [10] which are probably less prone to desiccation than herbaceous plants. Consequently, nectar volume levels measured in *D. lindenii* flowers were noticeably less in 2020 during a prolonged dry period compared to nectar levels measured the year before when conditions were wetter (J. Lu, M. LaRusso, unpub. data). Thus, it is conceivable that higher moisture levels could result in higher nectar levels in the floral spur. Considering that these other hawk moths species have proboscis lengths shorter than the mean nectar spur length [10], higher nectar levels may allow *D. lindenii* to be pollinated by a more diverse assemblage of hawk moth species, not just those with the longest proboscis lengths (e.g., *Cocytius antaeus*). This could result in more flowers being cross-pollinated between genetically diverse orchids leading to higher fruit set and higher embryo viability.

The record rainfall that occurred in the FPNWR in 2017 (264 cm) resulted in a prolonged hydroperiod extending into 2018 providing a glimpse of the conditions in south Florida during the 1930s and 1940s. For example, at Population A during the spring of 2018, we documented the largest wading bird rookery on record for the FPNWR, and this phenomenon was also noted on other conservation lands throughout the region. These observations coincided with a spike in the number of ghost orchid inflorescences being recorded in 2017 and 2018, as well as the number of seed capsules produced, which would have been the result of successful pollination by hawk moths. Taken together, we hypothesize that restoration of water to former levels, specifically unimpeded movement from Okaloachoochee Slough and East Hinson Marsh [19,20] into Fakahatchee Strand to the south, has the potential to benefit the fecundity of *D. lindenii*. Evidence from long-term hydrology data spanning the greater Everglades region is needed to fully assess freshwater flow patterns, and the extent that high water levels have changed over time. An analysis of historical data should also be included to gain a more complete picture. At the same time, more surveys are needed to fully assess the ecological drivers of population growth in the FPNWR and other populations throughout south Florida in future years.

Considering the widespread cypress logging coupled with water drainage during the last century, *D. lindenii* and its habitats that exist today likely consist of relic populations that persist in a state of very slow recovery. Although the life expectancy of the ghost orchid has yet to be determined under natural conditions, mature individuals are known to live for decades, many of which flower annually (E. Mujica, M. Danaher, pers. obs.). Thus, it is conceivable that the larger *D. lindenii* individuals observed in south Florida today were spawned from genetically diverse parents that were cross-pollinated by hawk moths on a grand scale prior to logging. Given that ca. 2000 ghost orchids are thought to remain in the region, about one-quarter of which are confined to the FPNWR (current study), the persistence of these elder sexually mature plants is crucial for repopulating existing sites and new sites with spontaneous seedlings. As our survey has revealed, however, mature plants are currently in an active state of decline and seedling recruitment is not expected to keep pace with the decline during the next decade. This negative trend may be exacerbated by poaching, climate change, and other factors. Restoring the hydrology, however, may be one way to increase orchid numbers through seedling recruitment as a way to offset some of the projected decline. Clearly, more data are needed in the years ahead to make more accurate population matrix projections. Special attention should also be given to identifying the variables that are linked to a particular cause or consequence and analyzing differences in supervised versus unsupervised data analysis (see [21]).

### 3.2. Future Directions

The timing of our survey coincides with the successful artificial propagation of *D. lindenii* from seed, and the successful outplanting of seedlings in situ and ex situ with modest survival. Hoang et al. [1] yielded a protocol that documented symbiotic and asymbiotic germination of the ghost orchid leading to the acclimatization of deflasked seedlings ex vitro [22,23]. Hundreds of these seedlings were subsequently reintroduced in situ at the FPNWR and ex situ at the Naples Botanical Garden for education and public display. The long-term survivability of these seedlings will be monitored to determine the degree of success achieved. Collectively, *D. lindenii* is poised for immediate recovery under the framework of integrated conservation [24] which could potentially be coordinated through a network of experts and institutions linked through the North American Orchid Conservation Center. While our studies in the FPNWR are projected to continue, we recommend that other ghost orchid sites throughout south Florida be surveyed (e.g., Fakahatchee Strand State Preserve, Corkscrew Sanctuary, Big Cypress National Preserve, tribal lands) following a common protocol. Special attention should be given to the role of hydrology in areas where ghost orchids occur naturally to determine if there is a link between water levels and fecundity. To gain a more complete picture of *D. lindenii*’s status throughout its range, plans are underway to compare the data from populations in Florida with those in western Cuba. As these combined activities are simultaneously carried out, we urge land managers and researchers to consider identifying potential new *D. lindenii* habitats on higher ground in the wake of rising sea levels imposed by climate change.

## 4. Materials and Methods

### 4.1. Study Site

The Florida Panther National Wildlife Refuge (FPNWR), part of the Big Cypress-basin Ecoregion, was chosen because it encompasses the northern portion of the Fakahatchee Strand known to harbor the highest density of ghost orchids, and because the orchid sites are remote and closed to the public. Within this 10,684 ha area are 27 orchid species in 17 genera [15], most of which are epiphytes confined to “islands” of strand swamps and sloughs shaded by bald cypress, *Taxodium distichum* (L.) Rich. (Cupressaceae), which constitutes the upper canopy. The majority of the ghost orchids are affixed to bark of trunks and branches of trees in the understory consisting of pop ash, *Fraxinus caroliniana* Mill. (Oleaceae), and pond apple, *Annona glabra* L. (Annonaceae). These trees are rooted in tannin-rich pools of standing water present most of the year except during dry periods (November-March). Epiphytic orchid associates of *D. lindenii* include three *Epidendrum* species (*E. amphistomum* A. Richard; *E. nocturnum* Jacq., *E. rigidum* Jacq.), *Polystachya concreta* (Jacq.) Garay & Sweet, *Prosthechea cochleata* (L.) W.E. Higgins var. ‘*triandra*’ (Ames) W.E. Higgins, and two other leafless species, *Campylocentrum pachyrrhizum* (Reich. *f*.) Rolfe, and *Dendrophylax porrectus* (Reich. f.) Carlsward & Whitten (syn = *Harrisella porrecta*) all of which are listed as “endangered” in Florida [5].

### 4.2. Field Sampling

Sampling was carried out during the month of July during six consecutive years (2015–2020). Prior to the sampling, potential ghost orchid habitats were identified by aerial photographs that revealed sloughs and ponds harboring standing water. These sites were then visited on foot to assess the presence or absence of *D. lindenii*. Habitats that did not have standing water between these sites were also inspected, but these areas were ruled out because they altogether lacked *D. lindenii* and other orchid epiphytes with one exception (*Encyclia tampensis*). All newly-documented ghost orchids were assumed to be present the previous year, given the notoriously slow growth of this species [25]. All living ghost orchids known to occur within the FPNWR were documented annually, including newly-discovered individuals. For modeling purposes, we limited our analysis in the yearly surveys to 409 individuals (2020 survey) to ensure consistency.

To locate each orchid annually, the host tree was plotted, tagged, and its GPS coordinates entered. Clusters of orchids on a given host tree were then individually marked using detailed distribution data at the point of discovery. For each orchid, we attempted to determine survival size and fruiting status. *Dendrophylax lindenii* is almost entirely leafless throughout its life, plant size was defined by the length of living roots and was a good predictor of vital rates as reported previously for the species [26]. Thus, root length served as the basis for our stated variable in the analysis. Root growth data were restricted to ghost orchids measured the previous year, not newly-discovered individuals. For reproductive output, the fruit number was used in our estimates.

The orchids were clustered in seven different populations separated by ca. 5 km from one another in moist strand swamps and seasonally flooded isolated wetlands. These populations are designated herein as Population A, B, C, D, E, F, and G. Quantitative data were collected for individual orchids and their host tree substrates. Host trees were numbered and identified, also noting information pertaining to the bark and the diameter at breast height (DBH) for each individual substrate on which a ghost orchid was found. Data collection for individual *D*. *lindenii* included plant height (i.e., distance above the soil), and DBH of the host tree, orientation, position on the tree (trunk/branch), flowering/fruiting status, and also information pertaining to the roots. Although ghost orchids are typically found on the main trunk of each host tree and below 3 m, branches higher in the canopy were also visually inspected. Root analysis included dead and living root counts, average living root length, and individual root lengths for each *D. lindenii*. For each orchid, the sum total of all root lengths combined was also calculated and was termed a “root string”. Individuals having a mass of roots that were unable to be quantified (height, aggregated, etc.) were recorded as outliers and were considered as mature plants. Seedlings (C-1 class) were defined as having 1–3 living roots with a mean root length of ≤3 cm, whereas juveniles (C-2 class) had 2–6 living roots with a mean root length ≤7.5 cm with ≤1 dead root, and no signs of an inflorescence in development. Mature plants were those with mean root lengths ≥3.5 cm and ≥2 living roots, with at least one root ≥10 cm in length, with smaller mature orchids grouped as class C-3, and the largest as C-4. Flowering individuals were recorded as having either an existing flower at the time of sampling, an inflorescence in development, or signs of an inflorescence from the previous year. To ensure accuracy, the previous year’s inflorescence was detached and discarded to exclude it from being counted the following year.

### 4.3. Vital Rate Estimation

The vital rate estimation values were based on previous work by Raventós et al. [26] for the *D. lindenii* population in western Cuba (Guanahacabibes National Park). Briefly, mean root lengths were used for the construction of integral projection models (IPMs; [27,28,29] to estimate *D. lindenii*’s vital rates as outlined in Raventós et al. [26]. This system also mirrors parameterizing matrix population models used by other authors (e.g., [30,31]. Five vital rate functions were calculated: (1) annual survival, (2) mean size after one year (i.e., mean growth), (3) residual variance in size (i.e., variance in growth), (4) probability of reproducing, and (5) number of fruits produced (conditional on reproduction). For each of the vital rate functions 1–4, we constructed a series of six models, including size, size^2^, categorical year effects, and size by year interactions. We then used logistic regressions for survival and for the probability of reproduction, and linear regressions were applied to the other two vital rates. AICc was used to identify the best model for each variable which was subsequently applied to parameterize the year-specific transition kernels for an IPM model. The distribution of recruit sizes when first observed (averaged across all years of data) was used in the calculation of the distribution of plant sizes the first year of age.

### 4.4. Demographic Modeling

To estimate temporal variability in demographic rates, size-structured IPMs were constructed for each annual transition (e.g., 2015–2016, 2016–2017). This was carried out by assembling the year-specific survival, growth, and reproductive rates into four separate matrices of 45 size categories of 1 cm width each, and each was analyzed as a separate time-invariant model. Similarly, we constructed a matrix of mean vital rates averaged across years. To verify that 45 size classes were indeed sufficient to yield stable predictions [28], mean matrices were constructed with 5–500 classes. This revealed that models with ≥10 classes yielded the same stable values of lambda (λ) to the 3rd decimal point. As category number was increased, this did not result in a trend in growth rate estimates.

In these models, each element *a_ij_* represented the sum of two products—one that includes growth and survival probabilities of an orchid, and the other representing its reproduction: *a_ij_* = *s_j_* ∗ *g_ij_* + pf*_j_* ∗ flr*_j_* ∗ rec ∗ rsize*_i_*, where *s_j_* = survival rate of the orchid in class size *j* (evaluated at the midpoint size for the class) and *g_ij_* = the normalized probability that a surviving plant will move from size *j* to size *i*. The second term is indicative of reproduction, where pfj = the probability of reproducing flrj = the mean fruit number produced by a reproducing plant of size j (both evaluated at the midpoint size of the class), rec = the recruit number per fruit, and rsize*_i_* = the probability that a seedling will reach a size of i.

The XToolsPro kernel density tool in ArcGIS Desktop 10.7.1 was used to create Figure 3. For point features marking orchid locations, the program was used to calculate the density of *D. lindenii* points in a neighborhood around those points. The kernel function was based on the quartic kernel function described in Silverman [32].

## Figures and Tables

**Figure 1 plants-10-01579-f001:**
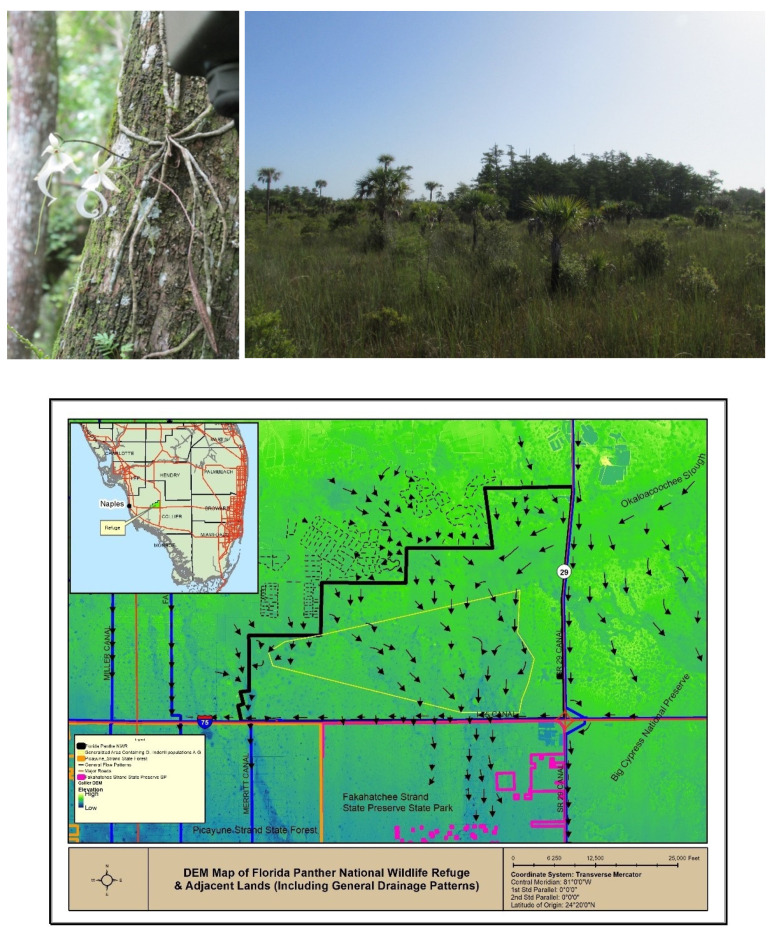
Left: *Dendrophylax lindenii* within the Florida Panther National Wildlife Refuge (FPNWR). Photo by M. Danaher. Right: The habitat of *Dendrophylax lindenii* within the Florida Panther National Wildlife Refuge showing an isolated cypress dome (background) where hydrology is conducive to supporting this species and other epiphytic orchids. The dome is surrounded by wetland prairie habitat dominated by sawgrass and frequented by cabbage palms (*Sabal palmetto*). Photo by L. Zettler. Bottom: Map of south Florida, USA showing the location of the study site (FPNWR). Most of the estimated 2000 remaining ghost orchids are confined to this region, especially within the FPNWR and Fakahatchee Strand Preserve State Park to the south.

**Figure 2 plants-10-01579-f002:**
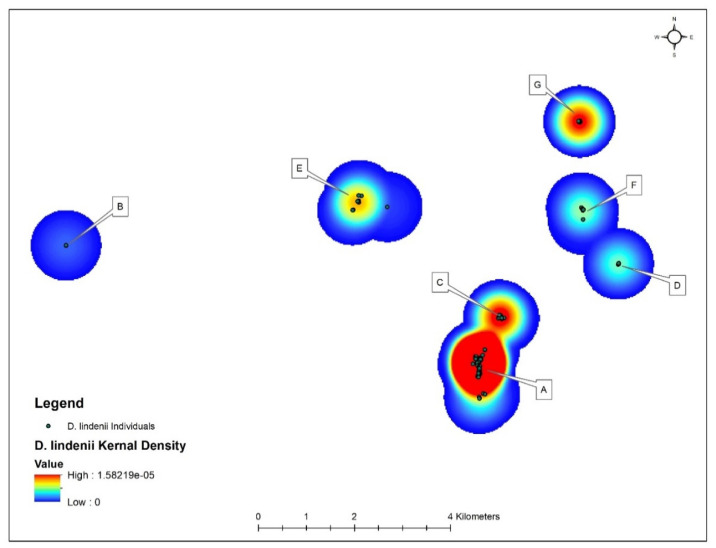
Visual depiction of the seven populations of *Dendrophylax lindenii* throughout the Florida Panther National Wildlife Refuge using kernel density analysis via ArcGIS. The colored circles depict orchid densities revealed as “hotspots”, with those in red indicating the highest densities. Note that Population A had the highest concentration of orchids and was the only site where spontaneous seedlings were recorded all 6 years.

**Figure 3 plants-10-01579-f003:**
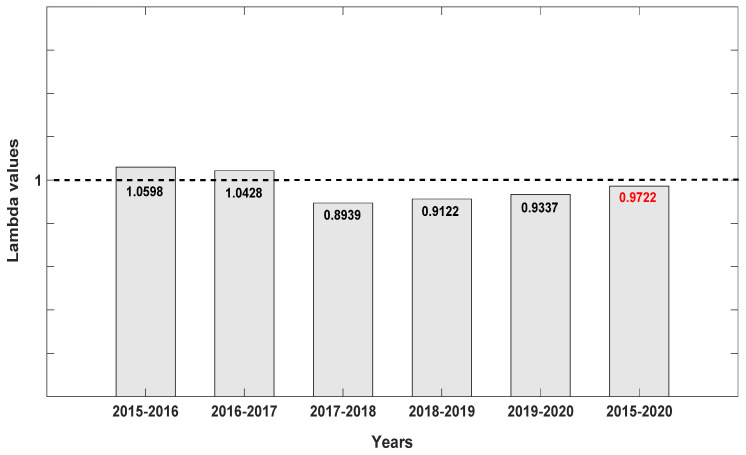
Lambda (λ) values for each pair of years and for the period 2015–2020 for all seven *Dendrophylax lindenii* populations within the Florida Panther National Wildlife Refuge. Note that the Lambda values (λ) indicate an increase in the first two years followed by a noticeable decrease (λ = 0.8939) in the third year (2017–2018). The figure shows a progression towards recovery during the last three years, but λ remains <1.

**Figure 4 plants-10-01579-f004:**
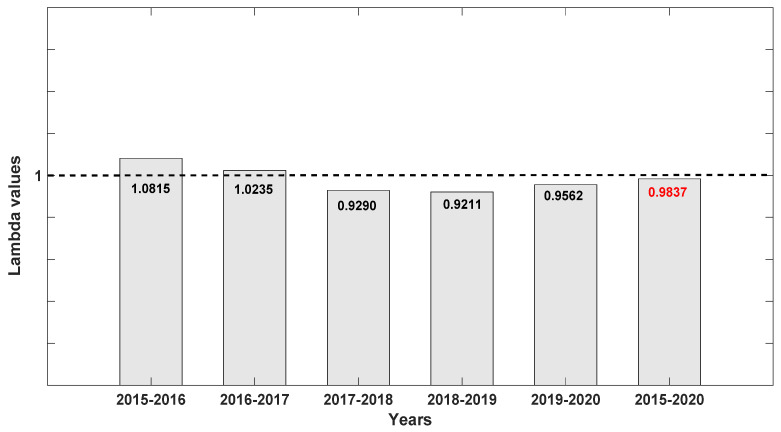
Lambda (λ) values for each pair of years for *Dendrophylax lindenii* in Population A. Note that these values are slightly higher for all six years except the second pair year (2016–2017), suggesting that Population A is most stable for now possibly because it harbored a high number of seedlings.

**Figure 5 plants-10-01579-f005:**
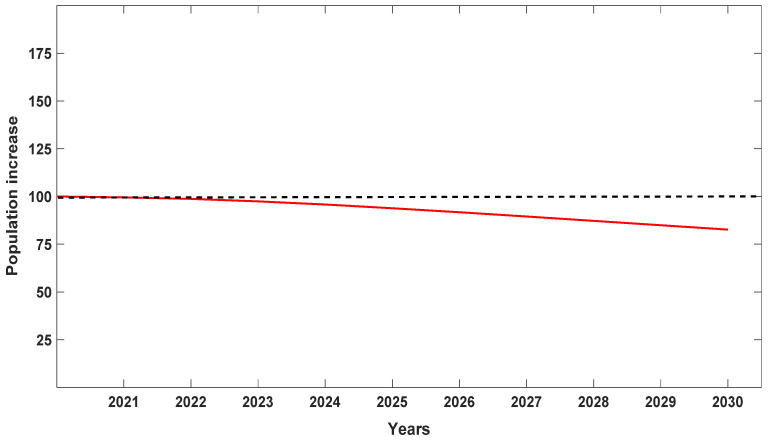
Projection of the behavior of *Dendrophylax lindenii* within the Florida Panther National Wildlife Refuge during the next 10 years for all seven populations combined. This graph suggests a 20% drop in ghost orchid numbers by 2030 in the absence of external adverse factors (e.g., hurricanes). If external factors are considered, the decline would be expected to accelerate.

**Figure 6 plants-10-01579-f006:**
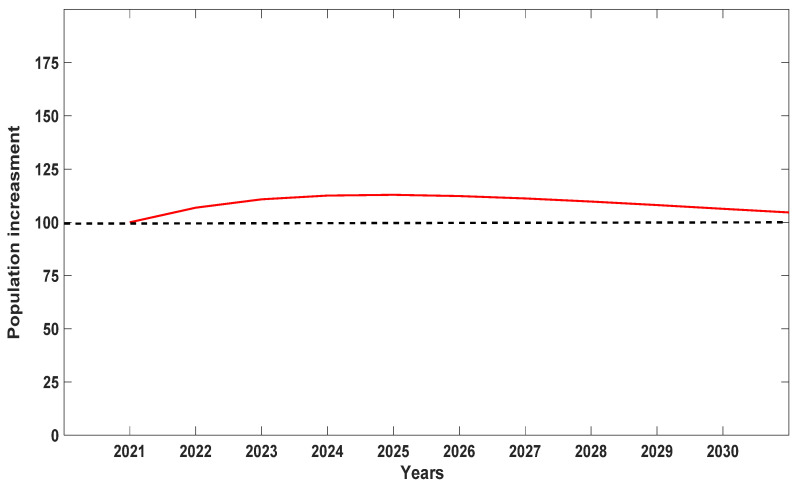
Projection of the behavior of *Dendrophylax lindenii* for Population A within the Florida Panther National Wildlife Refuge during the next 10 years. Compared to the other six populations, Population A is most stable because seedling recruitment is most evident at this site.

**Figure 7 plants-10-01579-f007:**
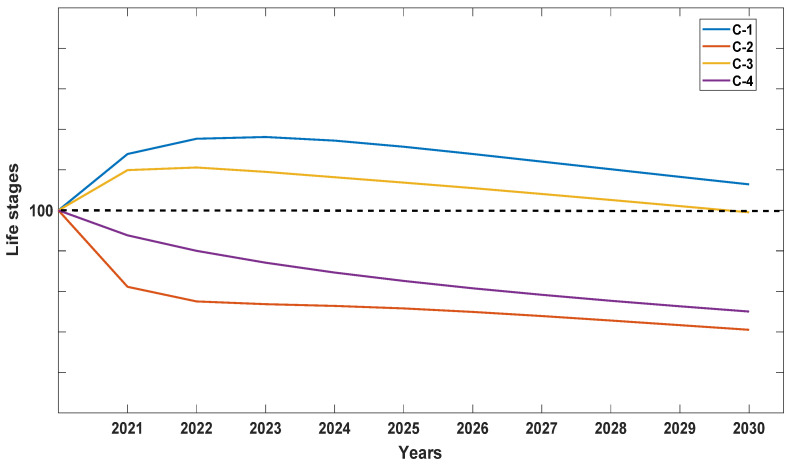
Projection of the behavior between the different orchid life stages (seedlings, juveniles, mature) for the next 10 years at Population A within the Florida Panther National Wildlife Refuge. Orchid life stages listed include seedlings (C-1), juveniles (C-2), smaller mature (C-3), and larger mature orchids (C-4). This graph shows that mature plants, like those in the other six populations, are in decline. These mature plants are the presumed source of the spontaneous seedlings observed in Population A. A drop in the number of mature plants would be expected to result in a drop in the number of spontaneous seedlings in the years ahead.

**Figure 8 plants-10-01579-f008:**
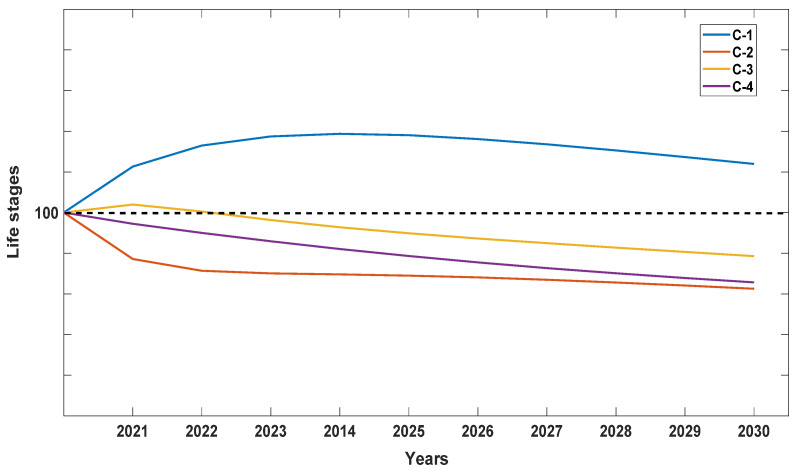
Projection of the behavior between the different orchid life stages (seedlings, juveniles, mature) for the next 10 years for all seven populations in the Florida Panther National Wildlife Refuge. Orchid life stages listed include seedlings (C-1), juveniles (C-2), smaller mature (C-3), and larger mature orchids (C-4). This graph shows that the number of mature plants (C-3, C-4) will decrease.

**Table 1 plants-10-01579-t001:** The total number of *Dendrophylax lindenii* individuals present in the Florida Panther National Wildlife Refuge (FPNWR) in a given year for each of the six years (Y1-6, 2015–2020) listed by growth stage and fecundity for all seven populations pooled. Numbers in parentheses reflect the percentage of the total each year. Orchid numbers listed for each year reflect those that were recorded as living at the time of the survey. This number includes those documented the previous year as well as newly-discovered individuals and excludes individuals that perished since the last survey.

-	Y1 (2015)	Y2 (2016)	Y3 (2017)	Y4 (2018)	Y5 (2019)	Y6 (2020)
Seedling	45 (24)	109 (44)	155 (44)	130 (32)	55 (14)	0 (0)
Juvenile	44 (23)	39 (16)	52 (15)	63 (16)	165 (41)	241 (59)
Mature	99 (53)	97 (40)	142 (41)	208 (52)	183 (45)	168 (41)
Total Orchids	188	245	349	401	403	409
Inflorescences	40	59	59	85	92	80
Flowers	56	101	78	123	96	82
Flowers/Inflor	1.4	1.7	1.3	1.4	1.0	1.0
Fruits	3 (5.4)	4 (4.0)	5 (6.4)	21 (17.1)	16 (16.7)	6 (7.3)

## Data Availability

Publicly available datasets were analyzed in this study.

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
