# Peer review of "Projected Status of the Ghost Orchid (Dendrophylax lindenii) in Florida during the Next Decade Based on Temporal Dynamic Studies Spanning Six Years"

_plants, 2021, doi:10.3390/plants10081579_

Round 1

Reviewer 1 Report

The authors described a multi-year monitoring study of the Ghost Orchid in Florida.  The species is of conservation importance and prior to this study there was no detailed information on the status of the species, especially the dynamics of known populations.  The authors identified a new population and provided the metrics, for the first time, to assess the status of exiting populations.  They found that seedlings were present and survived in only one population.  This result is alarming but also demonstrates that other studied are needed to confirm their hypothesis that the annual hydrology of hammocks is important. 

The writing is clear and I only offer a few suggestions (indicated as notes embedded in the text).

My only suggestion for improvement would be to provide more evidence from long-term hydrology data for the Everglades region that alteration of freshwater flow patterns, and the duration of flooding, have changed overtime.  Analysis of historical data would help support their hypothesis that there is now less flooding and for shorter duration - thus potentially having a negative effect of recruitment.

Author Response

We sincerely appreciate your constructive comments on our manuscript.  All of the edits have been incorporated (noted in gray shade), and we have included commentary regarding the need for long-term hydrology data (lines 289-295). 

Changes at the urging of the second reviewer are noted in blue shade, including the addition of the map in Figure1. 

During the course of the review process, the other co-authors and I made a few addition changes and these are noted in yellow shade.

Thank you again for your review of our manuscript.  

Sincerely, 

L. Zettler 

Reviewer 2 Report

Mujica et al. present an elegant and important study of the demographics of a rare orchid species. Without a doubt, this manuscript is scientifically sound and merits publication. However, the presentation lacks in a few minor aspects, that I advise the authors to address.

L 28: “from the wettest location”

L 29: I suggest avoiding the jargon “senile”. Simply write “most population within the FPNWR lack conditions…”

L 29-30: The last sentence should appear in L 27.

L 37: Does it have no leaves at all and at all life stages? If so, does it photosynthesize in the stem, or is it parasitic? This can be significant to understand its ecology and life cycle.

L 53-58: A small map will make a great difference (can be an additional panel in Figure 1).

L 68: As you well know, we need to know not only how many there are, but also how many there will be.

Table 1: This data is misleading. The total population did not increase by a factor of 10 from 2014 to 2015. You should note the number of new discovered individuals each year (preferably by stage).

L 117-118: Text does not fit table. I see a total of 188-409 each year, not 50-100.

Figures 3-4: I recommend merging these into one figure, showing lambdas of all populations next to lambda of Population A. This will enable readers to compare population A to the entire survey.

Figures 5-8: I recommend merging all these into a single figure with four panels. You may also consider having only to panels (Fig. 5+6, Fig. 7+8). It looks like a waste of space as it is now. In Fig. 7+8 I further recommend to change the graphical legends from codes to actual words: “seedlings” instead of “C-1”.

L 213-220: This justification should be in the Introduction. (L 220-223 can go either way)

L 238: If you insist on using the term “senile”, please define it in the same sentence.

L 249-251: You should probably mention (or even show) this the first time you mention Population A.

L 263: “OTU” (probably auto-correct did this, so be careful).

L 281-283: In Population A.

L 394-399: Is this classification based on literature?
